# A Dataset for Efforts Towards Achieving the Sustainable Development Goal of Safe Working Environments

**Eirik Lund Flogard**
Norwegian Labour Inspection Authority
Prinsens gate 1
Trondheim, Norway
eirik.flogard@arbeidstilsynet.no

**Ole Jakob Mengshoel**
Norwegian University of Science and Technology
Sem Sælands Vei 9
Trondheim, Norway
ole.j.mengshoel@ntnu.no

## Abstract

Among United Nations' 17 Sustainable Development Goals (SDGs), we highlight SDG 8 on Decent Work and Economic Growth. Specifically, we consider how to achieve subgoal 8.8, "protect labour rights and promote safe working environments for all workers [. . . ]", in light of poor health, safety and environment (HSE) conditions being a widespread problem at workplaces. In EU alone, it is estimated that more than 4000 deaths occur each year due to poor working conditions. To handle the problem and achieve SDG 8, governmental agencies conduct labour inspections and it is therefore essential that these are carried out efficiently. Current research suggests that machine learning (ML) can be used to improve labour inspections, for instance by selecting organisations for inspections more effectively. However, the research in this area is very limited, in part due to a lack of publicly available data. Consequently, we introduce a new dataset called the Labour Inspection Checklists Dataset (LICD), which we have made publicly available.[1] LICD consists of 63634 instances where each instance is an inspection conducted by the Norwegian Labour Inspection Authority. LICD has 577 features and labels. The dataset provides several ML research opportunities; we discuss two demonstration experiments. One experiment deals with the problem of selecting a relevant checklist for inspecting a given target organisation. The other experiment concerns the problem of predicting HSE violations, given a specific checklist and a target organisation. Our experimental results, while promising, suggest that achieving good ML classification performance is difficult for both problems. This motivates future research to improve ML performance, inspire other data analysis efforts, and ultimately achieve SDG 8.

## 1 Introduction

**Background.** Poor health, safety and environment (HSE) conditions in workplaces is a widespread problem that negatively affects both individuals and society. Every year in EU alone, more than three million workers are victims of serious accidents causing more than 4000 deaths due to poor working conditions [1]. World-wide, it has been estimated that at least 9.8 million people are in forced labour (2005) [2]. Labour inspections, a part of the International Labour Organization's "Decent Work Agenda," seek to preventing this and enforce regulations that protect workers' health, environment and safety [3]. Labour inspections are also important to globally achieve UN's Sustainable Development Goal (SDG) 8.8, "protect labour rights and promote safe working environments for all workers [. . . ]."

---

[1]The LICD dataset is located at `https://doi.org/10.18710/7U6TZP`.

36th Conference on Neural Information Processing Systems (NeurIPS 2022) Track on Datasets and Benchmarks.

Table 1: Summary of three inspections of three different organisations. The inspections are conducted in two different industries, with three different checklists. The last two rows show different checklists being used for similar organisations, illustrating that checklists are situation dependent.

| Checklist Content | Industry | County | Result |
|---|---|---|---|
| Working agreements, HSE and working environment training, Working hour schedules, Occupational health service, Building and equipment conditions, Risk assessment and measures, ... | Accommodation business | Oslo | Compliant |
| Market control - chemicals, Substance register. | Manufacture of metal products | Rogaland | Non-compliant |
| HSE and working environment training, Occupational health service, Mapping and risk assessment of chemicals and biological factors, Personal protective gear - technical, Noise exposure, ... | Manufacture of metal products | Viken | Non-compliant |

To identify poor HSE conditions, labour inspection agencies use checklists to survey organisations for non-compliance [4, 5, 6]. Each checklist contains questions that relate to common working environment violations within one or more industries. The answers to the questions indicate whether non-compliance to HSE and working environment regulations are found within the targeted organisations. When non-compliance is found, the agency follows up with reactions against the organisations. Checklists are also used in other high-stakes domains including aviation, food inspection and surgery [7, 8]. They are also used in machine learning (ML) for various tasks, such as testing and evaluating NLP models [9, 10] or fairness assessments [11].

Since labour inspection agencies usually have limited resources, it is vital that the inspections are carried out efficiently. This is challenging, as inspection strategies are difficult to adapt to a world that is constantly changing [12, 13]. Therefore, some agencies have recently started to use ML to select organisations for inspections [14, 15]. This is shown to increase inspection efficiency, as violations are far more common in the organisations that are selected using ML. However, ML research in this areas has been very limited. To our knowledge, and other than our previous work [15, 16], there is a lack of publicly available datasets to enable such research.

**The Labour Inspection Checklists Dataset.** In order to address SDG 8, specifically SDG 8.8, we present the Labour Inspection Checklists dataset (LICD) [17]. We aim to support and inspire ML research on labour inspections. Our current LICD dataset complements previous SDG-related NeurIPS datasets and benchmarks [18, 19], since none of them covers SDG 8. LICD is a Norwegian dataset, translated to English, that consists of results from inspections conducted by the Norwegian Labour Inspection Authority (NLIA) between January 2012 and June 2019. The dataset contains 63634 instances and 575 features. Each instance contains organisational and financial information about the organisation targeted for the inspection, a description of the checklist used and a binary variable that indicates whether violations were found. To demonstrate possible LICD use-cases, we introduce and show initial experimental ML results for these two problems:

***The Checklist Selection Problem (CLSP):*** *Let there be a selection of $N$ checklists and a target organisation* **x***. Given the $N$ possible checklists, select the best checklist $y$ to survey the target organisation* **x***. In this setting, the best checklist is the checklist that its user (the inspector) considers to be most relevant for surveying* **x***. The problem can be seen as a classification problem.*

***The Non-compliance Classification Problem (NCP):*** *Given a checklist $y$ and a target organisation* **x***, classify the target organisation's compliance $l$ to any of the regulations given by the content of $y$. The value of $l$ is unknown until the completion of the inspection and belongs to a Bernoulli distribution where $l = 1$ means that the target organisation is non-compliant and $l = 0$ means that the organisation is compliant.*

Both problems are non-trivial and important to promote safe workplaces. For CLSP, there are $N = 369$ different checklists that can be used for any given organisation. Table 1 demonstrates how labour inspection checklists can vary significantly, even among similar organisations. The reason for this is that organisations are subjected to numerous regulations and only a few of these are covered by each checklist. Selecting a checklist that covers all the most relevant subset of regulations is no trivial task, so CLSP is therefore difficult to solve. The task is also important, as a consequence of selecting wrong checklists may be that working environment violations are left unaddressed [15]. NCP is also important as it can be useful for selecting the best combinations of checklists and organisations for inspections, by predicting whether violations at a potential organisation is found when using a potential checklist. Alternatively, the checklist-component in NCP can be omitted so that the problem

only focuses on predicting violations in organisations [14]. Both CLSP and NCP could also be solved as a multi-objective problem of selecting the most relevant checklist (CLSP) that also maximizes the likelihood for finding non-compliance in the inspected organisation (NCP). However, for simplicity we treat CLSP and NCP as single-objective problems in this paper.

**Paper Overview.** The rest of the paper has the following structure. In Section 2 we present an overview of related work. A formal description of LICD is provided in Section 3. Section 4 describes how the data was collected and processed. An analysis of the dataset is provided in Section 5. Two baseline CLSP and NCP experiments are conducted in Section 6. In Section 7, we discuss implications of this work, including ideas for future work.

## 2 Related Work

**Regulation Enforcement Datasets.** Earlier, we used a dataset similar to LICD to construct new checklists via ML [15, 16]. However, the previous dataset only contains 4 independent features and cannot easily be used for other tasks beyond constructing new checklists. In contrast, LICD contains as many features as we have been able to collect (that can also be shared to the public within legal and ethical limits). Further, LICD is designed for checklist selection and non-compliance prediction rather than checklist construction. Since selecting existing checklists may be much simpler than creating new ones, an ML method for doing so may be easier to adopt for labour inspection agencies.

Superficially similar labour inspection datasets from other countries are publicly available, such as the Danish Smiley dataset [20]. Unfortunately, the Smiley dataset only contains instances where organisational non-compliance is found (positive labels); there are no instances where the organisation is compliant (negative labels) that also include inspection details. Another example is a dataset published by the American OSHA, which is frequently used in health and safety research [21, 22]. However, the dataset is not complete from an ML perspective since it only contains records about the regulations that were found non-compliant at the inspections (positive labels). The OSHA dataset also contains injury records, so Johnson et al. [23] propose to use ML on them to prioritize organisations for inspections. However, injury data may not be reliable for this purpose due to bias and under-reporting of accidents and injuries to authorities [24]. Other labour inspection datasets have been analysed [5, 6], but none are published openly. A mining safety inspection dataset is openly available [25], but such inspections are highly specialized; health and safety hazards and regulations in other industries are quite different. Datasets from other enforcement domains have also been published. These datasets include environmental inspections where ML has been used to predict violations in facilities [26]. There is also the Vancouver Crime dataset for law enforcement, which has been used for both criminology and ML research [27, 28]. Another example is the Chicago Food Inspection dataset, which includes an ML classification model [29, 30]. These datasets are fundamentally different from LICD, but highlight the importance of ML research within other branches of regulation enforcement.

**Checklists in Other Domains.** In addition to labour inspections, checklists are used in other situations where ensuring human health and safety is critical. In surgeries, checklists are used to ensure compliance to safety standards. For example, the WHO Surgical Safety Checklist from 2008 has substantial positive effects on patients' safety [31]. However, there are challenges related to implementation of the checklist in daily use. Some of the challenges are communication errors, lack of user compliance and lack of flexibility since standards of medicine varies from country to country [32]. Cockpit checklists in the aviation industry also face similar challenges, as improper checklist usage can lead to accidents [33]. The success of using a checklist may depend on having the correct content for a given context. Selecting the most relevant checklist for a given context is also one of the motivations for our work.

**Long-tail Classification.** In Section 5 we observe that the distribution of checklists in LICD is long-tailed and since the dataset contains $N = 369$ unique checklists, the dataset could be relevant for methods which address long tail classification problems. Dealing with long tail distributions, where classes tend to be distributed according to Zipf's law, can be challenging as models usually perform better when dealing with head-classes than the tail-classes [34]. Some well known long tail distributed datasets besides LICD already exist, such as CIFAR-100-LT, Fashion-MNIST or ImageNet-LT [35, 36], but these are not directly SDG-relevant. The most well-known approaches to deal with long tailed distributions are balancing methods, such as under- or oversampling [37]. Various ML and feature extraction-based methods to improve classification performance on long-

tailed datasets have also been proposed [38, 39, 40, 41]. However, the scope of this paper is to describe LICD rather than developing new methods for addressing problems such as classification of long-tail distributions.

## 3 Dataset Description

The LICD consists of 63634 instances with past inspections conducted by NLIA between 01/01/2012 and 01/06/2019 [17]. Each instance in LICD is described via 575 features and two target variables. Each feature represents either organisational or financial information about the inspected organisation. The first target variable is an identifier for the checklist that was used to inspect the organisation. The second target variable denotes whether non-compliance was found at the inspection. The features, target variables and column names of the dataset have been translated from Norwegian to English.

**Data Collection.** Let $\mathbf{D}$ be LICD (table) where each instance (row) consists of an inspection $\mathbf{d}$. The initiation of an inspection usually happens as a result of risk assessments. Each inspection takes 1-4 hours and is carried out using a checklist ($y$), which is a form that consists of yes/no questions. A no-answer to any of the questions means that the organisation is non-compliant ($l = 1$) to one of the regulations enforced by the agency. After the completion of the inspection, the completed checklist is uploaded digitally to NLIA's case management system. A report is then generated and delivered to the target organisation, regardless of the findings at the inspection. For any violations that are found, the corresponding questions and answers in the completed checklist are quoted in the report.

The information from each report is used to create an instance $\mathbf{d}$ in the dataset $\mathbf{D}$. Thus, each instance in the dataset is collected organically as part of NLIA's daily operations without additional interventions. Figure 1 shows the relations between various entities of the dataset. A description of these follows below.

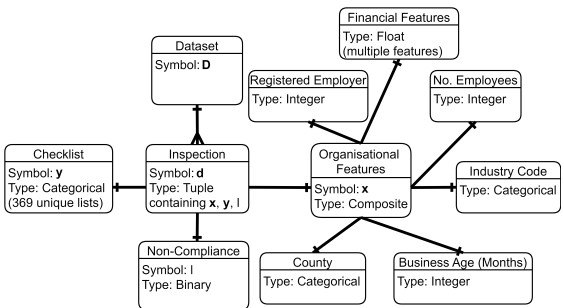

Figure 1: Diagram showing relations between different features and entities in the dataset. All of these are contained within a single table.

**Inspection.** An inspection is a single instance in LICD and can be viewed as a tuple $\mathbf{d} = (\mathbf{x}, y, l)$ where $\mathbf{x}$ is a target organisation, $y$ is a checklist and $l$ is the outcome of the inspection after using $y$ to survey $\mathbf{x}$. $\mathbf{x}$ consists of the 575 features in the dataset, while $y$ and $l$ represent the target variables.

**Organisation.** An organisation $\mathbf{x} \in \mathbf{d}$ is described by the 575 features in LICD. Each feature contains either organisational or financial information about the target organisation. Figure 1 shows how organisational information is contained in these features: Industry codes[2] (categorical), presence in the employer and VAT register (binary), county (categorical) and number of employees (integer). The other features, which are real numbers, contain financial information about the inspected organisation. Many of the instances in LICD have missing values ("NULL") for the financial features whenever these are irrelevant for the organisations' daily operations or fiscal reports. We therefore recommend replacing these missing values with 0 for the purpose of training ML models on the dataset.

**Checklist.** Each instance $\mathbf{d}$ in LICD contains a checklist $y$, used to survey an organisation $\mathbf{x}$. The content of $y$ is described by a column named Checklist Content, which lists of all the topics that are being investigated during the inspection. Every checklist $y$ is also associated with an identifier (Checklist ID). The Checklist ID is categorical and is one of the target variables in LICD. Since there are 369 unique checklists in the dataset, the Checklist ID can take on 369 different values.

**Non-Compliance.** Each inspection has an outcome $l$ in LICD, where $l = 1$ denotes that at least one of the questions on the checklist were found to be non-compliant (violation) at the inspection. The value $l = 0$ means that no checklist questions were found to be non-compliant. The outcome $l$ is also considered to be one of the potential target variables in the dataset.

---

[2]Interpretation table for industry codes: `https://www.ssb.no/en/klass/klassifikasjoner/6`

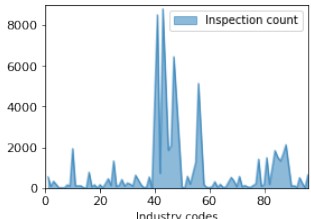
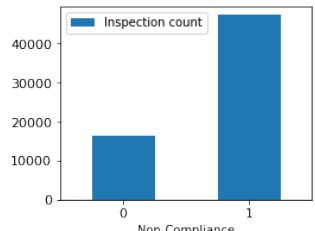
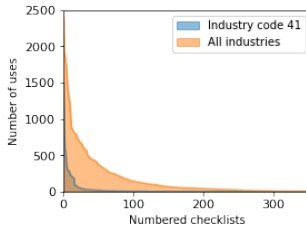

(a) Histogram of inspections over industry codes.

(b) Distribution of non-compliance in LICD.

(c) Histogram of checklists. Each unique checklist is denoted as a number on the x-axis.

Figure 2: Histograms of inspections, non-compliance and checklists with discrete unit bins on the horizontal axes. The vertical axes on the figures represent the number of occurrences in LICD.

## 4 Dataset Acquisition

**Processing and Validation.** LICD is a new dataset that is now being made publicly available, with exclusive permission from NLIA. The dataset was retrieved from the agency's databases using MSSQL17. Quality and integrity assurance is dealt with by NLIA's case management system, since this is essential for the system's operations (see Section 3). This assurance includes validation of data type, range and constraints. Consistency is also ensured by saving "information snapshots" of the target organisations at the time of the inspection, so that inspection records are unaffected by any updates to the organisations' registered information.

**Data Source Availability and Harm Prevention.** LICD contains information from sources that are publicly available, to some extent. Some of the organisational and financial features are available through Norway's official Central Coordinating Register, but the register has no historical records. The checklists and inspection outcomes are only available from inspection reports. Some of these may be available to the public, but access is granted only on a case-by-case basis via requests.

Since it is difficult to prevent identification of organisations in LICD, our strategy for harm prevention is to ensure that the dataset only contains information that is safe to make publicly accessible. For this reason we have not included details such as which regulations were found to be non-compliant. As an extra precaution we also deliberately make it difficult to identify organisations in the dataset. To do so we have excluded the organisation names, identifiers and precise locations from the dataset.

## 5 Analysis of the Dataset

In this section we conduct some analyses on LICD to highlight how the target labels and some of the key features of the dataset are distributed. We focus on industry and location (county), because these features are known to be important for labour inspections [12, 15, 16].

**Distribution of Inspections Over Industries.** Labour inspections are industry-oriented [12], so Figure 2a shows a histogram of inspections over different industry codes in the dataset. The horizontal axis represents the industry codes, which can be regarded as an ordinal feature where each number represents a specific industry. Industries with codes in close proximity to each other are often related. The vertical axis shows how many inspections that have been carried out in a particular industry code. As seen on the figure, most of the inspections are focused on industry codes from 50 to 60, which correspond to most of the building and road construction industries.

**Distribution of Non-Compliance.** As seen in Figure 2b, non-compliance ($l = 1$) is found in more than 40000 inspections in LICD. In other words, at least one violation is found in 74% of the inspections in the dataset. The fact that the majority of the inspected organisations are found to be non-compliant reflects the importance of labour inspections, in terms of correcting health, environment and safety problems in workplaces.

**Distribution of Checklists.** Figure 2c shows a histogram of the observation count of each unique checklist for all industries. The checklists are shown on the horizontal axis, ordered according to their number of observations in LICD. The vertical axis shows the number of observations in the dataset. The figure suggests that checklists usage follows a long-tailed distribution [34], since only a small fraction of the checklists are used very often.

Figure 2c also shows the distribution of checklists used within industry code 41, which is long-tailed. Industry code 41 corresponds to "building construction" and is one of the industries with the highest number of inspections in LICD. The figure reveals that more than 140 different unique checklists are used for inspections within that industry. However, most of the inspections are carried out using 20 of the available checklists. The high number of checklists used within a single industry code points to the fact that there is a significant diversity in the health, environmental and safety risks for organisations, even within the same industry.

**Distribution of Inspections Over Regions.** Table 2 shows the distribution of inspections across different counties or regions (fylke). Most inspections are carried out in Viken and Oslo, the two regions with the highest populations in Norway. The inspection counts seem to correlate with the population count of each region.

Table 2: Overview of the distribution of inspections for different locations in the dataset.

| County | Count # |
|---|---|
| Agder | 3375 |
| Innlandet | 5179 |
| Møre & Romsdal | 4368 |
| Nordland | 3555 |
| Oslo | 7688 |
| Rogaland | 5358 |
| Svalbard | 59 |
| Troms & Finnmark | 4148 |
| Trøndelag | 5463 |
| Vestfold & Telemark | 4922 |
| Vestland | 5773 |
| Viken | 13746 |

# 6 Demonstration Experiments

In this section we conduct two simple experiments to demonstrate potential use-cases for LICD. The first experiment addresses the problem of predicting the best checklist for a given organisation. The second experiment addresses the problem of predicting non-compliance, when a specific organisation and checklist is given. Due to the high number of features in LICD, we also evaluate some simple feature selection algorithms. Feature selection is known to promote explainability for ML in tasks involving high-stakes decision making, such as labour inspections [42, 15]. Feature selection can also improve computational and model performance via data dimensionality reductions [43, 44] and could therefore be another use-case for LICD. We used an earlier, unpublished version of LICD to evaluate feature selection methods in our previous work [43].

## 6.1 Data Preprocessing

For the experiments, we use one-hot encoding for categorical features. The other features are used directly in their original form. Many of the financial features contain missing values, which are denoted as NULL (see Section 3). We replaced these missing values with 0 for the experiments, because this is usually the most correct interpretation. We also excluded financial features with less than 10 observations, since these are unlikely to be useful for ML. Since we are using feature selection for the experiments, we decided to set this threshold fairly low.

## 6.2 Setup and Environment

The feature selection and ML methods used for the experiments are implemented via Scikit-learn. We study the following methods for feature selection: Anova F, $\chi^2$, Model Coefficients, Mutual Information, Forward Selection and Recursive Elimination. The ML methods used for the experiments are: Decision tree (DT), Logistic regression (LR), Naive Bayes' Classifier (NBC), K-Nearest-Neighbor ($k$-NN), AdaBoost, GradientBoost and Multi layered Perceptron (MLP). We are using GridSearchCV for hyper parameter tuning for $k$-NN, AdaBoost, GradientBoost and MLP. For each method, prediction threshold is set to 0.5 (NCP) or the class that has the highest prediction score (CLSP). We decided to not set the prediction threshold individually for each method, in order to keep the experiment simple and to avoid introducing bias.

Each ML method is always evaluated on 8 different feature set sizes that are selected via feature selection. The set sizes are 0.1%, 0.5%, 1%, 5%, 10%, 20%, 30%, 40% and 50% of the 575 features in the dataset, rounded up to the closest integer.

A Dell Precision 5560 laptop with Intel i9 11950h at 5Ghz, 64GB RAM at 3200Mhz, Nvidia Quadro RTX A2000 and Windows 10 are used for the experiments. The experiments are conducted in a Python environment using Scikit-learn 0.24 and Jupyter Notebook.

Table 3: Prediction performance with average standard deviations and run times from the CLSP main experiment on LICD. Times are measured in seconds.

| Method | Mutual Info | | | | Anova F | | | | Time |
|---|---|---|---|---|---|---|---|---|---|
| | Bal. Acc | Acc | Prec | Rec | Bal. Acc | Acc | Prec | Rec | |
| LR | .01±0 | .02±.01 | 0±0 | 0±0 | .01±0 | .03±.01 | 0±0 | .01±0 | 381 |
| NBC | .01±0 | .01±0 | .01±0 | .01±0 | .02±0 | .01±.01 | .01±0 | .02±0 | **13.6** |
| DT | **.06±.01** | **.09±.01** | **.05±0** | **.04±.01** | **.06±.01** | **.09±.02** | **.05±0** | **.05±.01** | 33.2 |
| k-NN | .05±.01 | .08±.01 | **.05±.01** | .03±0 | .04±0 | .07±.01 | .04±0 | .03±0 | 40.0 |
| AdaBoost | .01±0 | .05±.02 | 0±0 | .01±0 | .02±0 | .04±.02 | 0±0 | .02±0 | 619 |
| GradientBoost | .03±.01 | .06±.02 | .03±.01 | .02±.01 | .04±.01 | .08±.02 | .03±0 | .03±0 | 22316 |
| MLP | .01±.01 | .04±.02 | .01±.01 | .01±0 | .02±.01 | .05±.02 | .02±.01 | .02±.01 | 338 |

## 6.3 Experiment 1: The Checklist Selection Problem (CLSP)

The goal of this demonstration experiment is to establish baselines for solving the CLSP problem described in Section 1. The experiment is broken down in two phases: a pilot experiment where we evaluate feature selection methods for the problem, and the main demonstration experiment.

**Evaluation of Feature Selection Methods.** Since LICD contains many features, we decided to assess a range of feature selection methods for the main experiment. The feature selection methods and evaluation details are discussed in Section 6.2. We use DT for the evaluation, since it is fast and compatible with all the feature selection methods. After assessing each feature selection method, we then take the top three best performing methods in terms of accuracy and use the two fastest performing methods for our experiment.

The results from testing the feature selection methods are listed in Table 4. There are only minor differences between most of the methods in the test. Forward Selection, Anova F and Mutual Information have the best recorded accuracy scores. Forward Selection has the highest accuracy but the score is recorded from one run where only 0.1% of the features are selected. Forward Selection was unable to complete within two hours for any of the larger feature sets. We decided to move forward with Anova F and Mutual Information for the main experiment, since they are the fastest performing methods among the top three with highest accuracy.

Table 4: Result from the feature selection evaluation for CLSP using DT, with time measured in seconds.

| Method | Acc | Time |
|---|---|---|
| Anova F | **.106** | **.75** |
| $\chi^2$ | .070 | .78 |
| Model coefficients | .099 | 19.1 |
| Mutual Info | **.106** | 296 |
| Forward Selection | **.108** | 6146 |
| Recursive elimination | .105 | 306 |

**Design of the Main Experiment.** We are using the ML methods and setup described in Section 6.2. Each feature selection algorithm is applied to LICD before model training and evaluation, using the 8 configurations of feature set sizes described in Section 6.2.

After performing hyper parameter tuning, the optimal configuration for AdaBoost is $0.5$ learning rate and $50$ estimators. The optimal configuration for GradientBoost is $0.1$ learning rate and $20$ estimators. For $k$-NN, the optimal configuration is distance-based weights and $k = 100$ neighbors. The best settings for MLP are logistic activation function, 100 hidden layers and constant learning rate.

Each configuration is evaluated using 5-fold cross-validation with randomly selected training-evaluation sets from LICD. The performance is measured in terms of balanced accuracy[3], accuracy, precision and recall scores using the available methods in the Scikit-learn library. The average standard deviation for each cross-validation is also recorded. The results for each method are recorded by calculating the average of the scores reported from the 8 feature selection configurations.

**Results and Discussion.** The results from the main demonstration experiment are shown in Table 3, where the best score in each column is highlighted. In overall, it seems that achieving high prediction performance for CLSP is challenging. The prediction performance scores are reasonably low since the target variable consists of 369 different classes. When comparing accuracy and balanced accuracy in the table, the accuracy is in most cases greater than the balanced accuracy. This is probably caused by the long tailed distribution of the target variable classes (unique checklists) in the dataset. Out of the ML-methods we tested, DT had the highest balanced accuracy, accuracy, precision and recall scores. The results are somewhat surprising, especially since both AdaBoost and GradientBoost

---

[3]Balanced accuracy is accuracy score calculated via Sklearn with class-balanced sample weights.

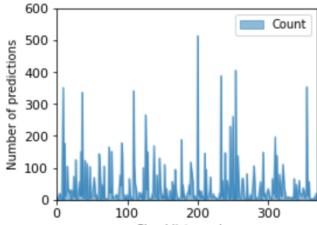

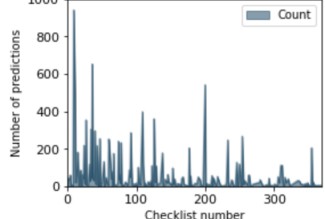

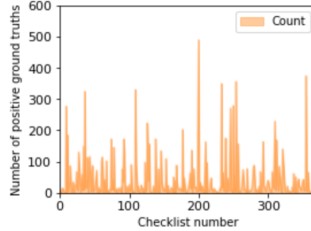

(a) Distribution of predictions for DT, using feature selection with Anova F and a 10% set size.

(b) Distribution of predictions for $k$-NN, using Mutual Info and a 10% feature set size.

(c) Distribution of ground truth labels.

Figure 3: Distributions on the evaluation set of a random paired 80-20 training-evaluation split. The horizontal axes represent the identifiers for 369 possible checklists (classes). The vertical axes on the figures represent the number of observations for each class in the evaluation set.

are based on using Decision trees as weak learners to improve model predictions. However, these methods are also susceptible to overfitting, which could be caused by the long-tailed class distribution of the target variable [34]. $k$-NN also performed well in comparison, ranked as second best on nearly all the scores in Table 3. However, the results show that there is not significant differences in balanced accuracy, accuracy and precision between the two best configurations of $k$-NN and DT. By naively predicting the most frequently used checklist from Figure 2c as positive, one can expect an accuracy and precision score of $\frac{2414}{63634} \approx 0.04$. Both $k$-NN and DT perform better than this, which is promising. It is still questionable whether these methods also would outperform an inspector or domain expert. Time-wise, DT has an average cross-validation time of 33.2 seconds. This is slightly better than $k$-NN's 40.0 seconds.

Figure 3a and 3b show the number predictions that each of the 369 checklists receives from DT and $k$-NN, and offer additional insights regarding the performance of the two best configurations in Table 3. Ideally, the distributions in Figure 3a and 3b should be similar to the distribution of ground truth labels in Figure 3c. Compared to the distribution of predictions from $k$-NN, the distribution for DT is more similar. In particular, the checklist with number 200 has the highest number of observations in both Figure 3a and 3c. For $k$-NN, checklist number 9 has 940 predictions which is over 3 times as many as the number of ground truth labels. Also, the predictions are much more concentrated between checklist number 0-100 in comparison to DT and the ground truths. Despite the similarities between Figure 3a and 3c, the amount of true positives (matches between predicted checklists and ground truths) for DT is low. One possible reason for this discrepancy is discussed in Section 6.5.

The overall difference in prediction performance between Mutual Information and Anova F in Table 3 is minor, but may slightly be in favour of Anova F for all the methods except $k$-NN and AdaBoost. Despite this, the highest ranked features for Mutual Info seem to be more relevant than for Anova F. The highest ranked features for Mutual Info include business age, public fees, payroll costs, industry codes and counties. For Anova F, the highest ranked features are business age and financial features such as assets, equity and bank deposits, dept, costs (including payroll), revenues, investments and profits. Industry codes are not ranked high by Anova F, which is surprising considering that checklists aim to address HSE risks which typically are industry-specific [15]. Thus, the results indicate that feature selection for ML is not always intuitive.

## 6.4 Experiment 2: The Non-Compliance Classification Problem (NCP)

The goal of this experiment is to solve the NCP problem described in Section 1. The experiment is structured in two phases in the same manner as in Section 6.3, with an evaluation of feature selection methods before the main experiment.

**Evaluation of Feature Selection Methods.** The assessment and selection of feature selection methods is done as in the previous experiment, where the two fastest performing methods out of the three methods with the highest accuracy are selected for the main experiment. We are using the feature selection methods and the feature set sizes described in Section 6.2. The feature selection methods are evaluated using DT.

Table 5: Result from the feature selection evaluation for NCP with DT.

| Method | Acc | Time |
|---|---|---|
| $\chi^2$ | **.667** | **.29** |
| Anova F | **.684** | .58 |
| Mutual Info | .661 | 229 |
| Model coefficients | .657 | 12.3 |
| Forward Selection | **.750** | 5464 |
| Recursive elimination | .658 | 300 |

Table 6: Results with average standard deviations from the NCP main experiment. The average time in seconds per cross-validation is shown on the far right column.

| Method | $\chi^2$ | | | | | *Anova F* | | | | | *Time* |
|---|---|---|---|---|---|---|---|---|---|---|---|
| | *Bal. Acc* | *Acc* | *Prec* | *Rec* | *AUC* | *Bal. Acc* | *Acc* | *Prec* | *Rec* | *AUC* | |
| LR | .42±.02 | .44±.02 | .68±.02 | .46±.02 | .41±.02 | .46±.02 | .45±.02 | .72±.02 | .43±.02 | .47±.02 | 3.56 |
| NBC | .56±.04 | .49±.11 | .72±.20 | .42±.18 | .57±.04 | .57±.02 | .53±.02 | .81±.02 | .48±.02 | .59±.02 | **1.06** |
| DT | .54±.01 | .45±.02 | .59±.01 | .35±.03 | .57±.01 | .51±0 | .30±0 | .20±0 | .08±.01 | .56±.01 | 14.2 |
| *k*-NN | **.58±.02** | **.53±.04** | .81±.01 | **.49±.07** | .61±.02 | .57±.02 | .52±.02 | .79±.06 | .47±.03 | .61±.02 | 100 |
| AdaBoost | **.58±.01** | .51±.03 | **.82±.01** | .44±.07 | **.63±.02** | **.62±.01** | **.57±.02** | **.84±.01** | **.52±.04** | **.68±.02** | 241 |
| GradBoost | **.58±.01** | .50±.03 | **.82±.01** | .43±.06 | **.63±.02** | **.62±.01** | **.57±.02** | **.84±.01** | .51±.04 | **.68±.02** | 1352 |
| MLP | .53±.01 | .40±.03 | .78±.02 | .27±.05 | .53±.01 | .54±.01 | .39±.02 | .82±.02 | .23±.05 | .56±.02 | 27.7 |

The results from the evaluation are shown in Table 5. The best performing method was Forward Selection, but we were only able to run it on the 0.1% and 0.5% feature set sizes within the time limit of two hours. Thus, the recorded accuracy is the average for only these two feature sets. The method with the second best accuracy score is Anova F and $\chi^2$ is the third best scoring method. Time-wise $\chi^2$ is the best performing method with an average completion time of 0.29 seconds for all the feature set sizes. The worst performing method was Forward Selection with almost 1.5 hours. Even though Forward Selection is the best performing method in terms of accuracy, the long running time makes it unfeasible for large feature sets or ML-methods with high computational complexity. Thus, we decide to use $\chi^2$ and Anova F for the main experiment.

**Design of the Main Experiment.** For the experiment, we use the ML methods and configurations described in Section 6.2. We also performed hyper parameter tuning for AdaBoost, GradientBoost, MLP and $k$-NN. After performing hyper parameter tuning, the optimal configuration for AdaBoost is 1.0 learning rate and 200 estimators. The optimal configuration for GradientBoost is 0.1 learning rate and 500 estimators. For MLP, the optimal configuration is logistic activation, 20 hidden layers, 0.0001 alpha and adaptive learning rate. The best configuration for $k$-NN is $k = 500$ neighbors and uniform weights.

All the ML methods in the experiment are evaluated using 5-fold cross validation where each fold consists of separate training (72%), test (8%) and evaluation sets (20%). Since the target variable is unbalanced with 74% positive labels (see Figure 2b), the prediction thresholds for the evaluation sets are set to the median prediction-scores of the corresponding test sets. As a result there should be an approximately equal number of predicted positives and predicted negatives for each evaluation set. The performance of each ML method is measured using the same statistics with standard deviations as in Section 6.3. Area under receiver operating characteristic curve (AUC) is also used.

**Results and Discussion.** The results from the experiment are shown in Table 6. These results are a bit more nuanced compared to the previous experiment. The best configurations are AdaBoost and GradientBoost with Anova F for feature selection, with significantly higher classification performance scores compared to $\chi^2$. For $\chi^2$, the best performing methods are AdaBoost, GradientBoost and $k$-NN. $k$-NN has slightly higher accuracy and recall scores than AdaBoost and GradientBoost, but AUC and precision scores are slightly lower. However, these differences are not significant. Time-wise, $k$-NN has the best performance with an average time of 100 seconds. AdaBoost outperforms GradientBoost with an average time of only 241 seconds, compared to 1352 for GradientBoost. Overall, AdaBoost seems to be the best method in this experiment, in terms of both time and classification performance.

The AUC score for AdaBoost is only 0.68, which indicates that there is room for improvements in terms of classification performance. On a first glance, the precision score of 0.84 looks better than the AUC score. However, the dataset is imbalanced 74:26 towards positives (non-compliance), so predicting all labels in the dataset as positive would yield a precision of 0.74. The difference is only 0.12 points, but this also translates into a 66% increase (3.16 to 5.25) in the odds of finding non-compliance in an organisation that is predicted as positive by the model. The difference is therefore an important improvement from a labour inspection perspective.

The highest ranked features from feature selection with Anova F are business age, number of employees, unpaid public fees, revenue, costs, industry codes and county (location). For $\chi^2$, the highest ranked features are somewhat limited in terms of variety and only include financial features related to taxes, revenue, costs, loans and equity. Thus, we would argue that Anova F selects a better,

more informed feature set compared to $\chi^2$. Anova F also yields higher performance than $\chi^2$ for most of the methods in Table 6.

## 6.5   Machine Learning Performance for CLSP and NCP

While some of the results from the experiments look promising, none of the methods we tested perform very well in terms of classification performance on either CLSP or NCP. This is also a motivation to use LICD in future research that aims to develop new, better performing ML methods. For this domain, a high precision score is arguably the most important statistic because it means that more effective checklists are being selected (CLSP) or a higher rate of non-compliance is found per inspection (NCP). A high recall is also desirable, but less important. Accuracy or AUC is also important, in order to ensure that ML methods have decent classification performance.

For CLSP, the highest recorded precision score in Table 3 is only 0.05 (DT). This is better than the naive selection strategy (0.04), but it is still questionable whether DT or any of the other methods perform well enough to be useful from a labour inspection perspective. However, it may be possible to improve performance by treating CLSP as a recommendation problem where a fixed number of checklists with the highest prediction scores are selected, leaving the user to decide which of them to use. This approach could be considered since the results in Figure 3 may suggest that there often are multiple optimal checklists for a given organisation, while CLSP assumes that there is only one. Collecting more information or features for LICD could be another way to improve performance, but more research is needed in order to understand what kind of information that should be collected.

There may also be ways to increase ML classification performance for NCP. The results for Forward Selection in Table 5 indicate that wrapper-based methods for feature selection may increase accuracy. Although using Forward Selection on LICD seems to be time-wise infeasible for feature sets larger than 0.1%, some of the more recent wrapper-based algorithms such as Stochastic Local Search or Differential Evolution could be more viable as they have comparably lower computational costs [43, 45, 46, 47].

## 7   Conclusion and Future Work

In this paper we have proposed LICD, a new dataset with labour inspection checklists. The dataset can be used to address working environment violations in organisations. Addressing such violations is important for efforts towards achieving SDG 8.8, to "protect labour rights and promote safe working environment for all workers". Research on ML for labour inspection is currently limited, so our motivation for this work is to promote further research on this subject.

LICD consists of 63634 instances with past inspections conducted by NLIA. The dataset contains 575 features and 2 possible target variables: Non-compliance and Checklist. Based on the target variables, the dataset could potentially be used for the following tasks: a) To select the most relevant checklist to inspect a given target organisation (CLSP). b) To classify whether non-compliance is found at an inspection, for a given organisation and checklist (NCP). Two demonstration experiments with CLSP and NCP suggest that they are promising but difficult problems, thus motivating future research.

A potential direction for future work is to explore ways to solve a combination of both CLSP and NCP as a multi-objective optimizations problem: to select the most relevant inspection checklist that maximizes the number of violations found in a given organization. A simple way to solve the problem could be to use a two-stage approach for selecting checklists. The first stage could be to select a subset of relevant checklists as candidates for the second stage. The second stage could then be to select the candidate that is most likely to be classified as non-compliant. Addressing a combination of CLSP and NCP could provide valuable decision support for inspectors and the approach can potentially be easier to adopt for inspection agencies, compared to our previous work where ML is used to create new checklists [16, 15]. Another direction is to develop more accurate ML methods for solving the CLSP and NCP problems, especially since none of the methods in our demonstration experiments achieved outstanding results in terms of accuracy, precision and recall scores. For the same reason, LICD could be relevant for benchmarking ML methods. The dataset could for instance be used in an SDG framework like SustainBench [18], which currently lacks a dataset that addresses the SDG on decent work and economic growth (SDG 8).

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
