# OpenReview forum: "A Dataset for Efforts Towards Achieving the Sustainable Development Goal of Safe Working Environments"
_NeurIPS.cc/2022/Track/Datasets_and_Benchmarks — NeurIPS 2022 Datasets and Benchmarks _

### Official Review · Reviewer_PyfR · 2022-07-15
**LICD dataset, possibly new?**

**Rating:** 5
**Confidence:** 4

**Strengths:**

The problem the paper attacks is an interesting one which has real, practical applications and could help workers.

**Weaknesses:**

While the dataset seems sound, some choices in the design of experimentation makes it so that the proposed baseline is weaker than it needs be, namely:
- For the checklist selection problem, it is not a 369 way classification problem, as not every checklist can be applied to every industry, which reduces the number of classes.
- The experiments used are simple statistical methods, which don't seem to have been selected with any particular reasoning behind them. Given the promises of DL in pretty much ever area, it feels like a severe limitation to not have at least a BERT or some pre-trained algorithm as a baseline.
- The feature selection method is done using a shortcut, where methods are evaluated only on DT and eliminated based on this result. Given the short running time of most of the algorithms, it would have been better to simply eliminate Sequential Selection and run every selection method on every algorithm. There’s also no report of the result without feature selection.
- The 5-cross validation seems like a good way to have results that people can't reproduce. The dataset is large enough to be split into a standard train/dev/test, which would allow practitioners to compare on the same test set. In the same vein, it is not clear how you fine-tune your algorithms given the absence of the test set. The methodology makes me believe that there is possibly peeking involved in the early stage (such as the feature selection)
- It would be beneficial to extend on the differences between the previously released dataset by the same authors (in 13, 14) and this one. One of the highlighted contribution is that this is a new dataset, but it seems like it's the same data with different preprocessing, which would make it new tasks on the data. While this doesn't change what is accomplished, it would bring more transparency to the paper.

**Additional Feedback:**

In the resubmission statement you state that you "translated" the dataset to English. This raises a bit of a concern as it is not mentionned in the paper.

**Clarity:**

A lot of acronyms are used, which can hinder the clarity of the paper. For example, it would be helpful to expand a bit on what SDG means since it feels central to the contribution of the paper. It would also be worthwhile to spend a little more time describing where the checklists come from.

**Correctness:**

The claims seem correct. Experiment design could be improved, as highlighted above.

**Documentation:**

The dataset is documented correctly

**Ethics:**

There is no ethic problem

**Relation To Prior Work:**

It would be helpful to expand on the difference between the previously released dataset in 13,14 and this one. Is it the same data?

**Summary And Contributions:**

The paper proposes a dataset of ~64K points for two tasks: selecting the best checklist for inspection and predicting non-compliance

---

> ### Author Response · Authors · 2022-08-09
> **The paper introduces a new data set (LICD) along with two new problems.**
>
> Hi,
> thank you for taking your time to review the paper. We have some comments regarding the weaknesses:
>
> - Although not every checklist is applicable to any industry, we would still argue for the classification problem approach as any checklist could in practice be selected for a given inspection. Any wrong predicted checklist that is inapplicable is simply be regarded as a misclassification or a wrong prediction if they don't match the ground truths.
>
> - Pretrained models like BERT are primarily used for natural language processing (NLP). However, none of the two classification problems are NLP tasks so we don't think it is possible to just apply one of these models to any of the problems we introduced in the paper. However, it may be possible to do NLP based tasks on the data set, but this would be outside the scope of our paper and would have to be addressed in future work.
>
> - The reason why we selected only one machine learning algorithm (and not many) for the initial feature selection test was to simplify the main experiment by reducing the number of feature selection methods. Using every feature selection method on every machine learning method could make the paper difficult to read. We also selected two feature selection methods for the main experiment (rather than one) based on the initial evaluation on DT, which we also believe that will reduce any potential bias towards DT. In one of the experiments, sequential selection also yielded some promising results in terms of ML performance. We have therefore updated Section 6 of the paper with a discussion of this.
>
> - Regarding not splitting data into training/test/evaluation:  We used Sklearn for all the machine learning methods we tested in the paper. For methods where it is necessary (like GradientBoosting), it automatically creates a test/dev-set from the training data (so that there are 3 sets in total) when relevant for fine-tuning the algorithm. Regarding the feature selection, we used only filter methods for the main experiments. These are used as a preprocessing step and doesn't rely on any optimization or tuning that would normally require splitting the data into 3 sets (in contrast to filter methods, wrapper methods are based on tuning and would require 3 sets). Since we don't do any explicit tuning in any of the experiments, we therefore see it as sufficient to use cross-validation with explicit training/evaluation sets (2 sets).
>
> - The data set described in [13, 14] is a different data set with different features, created specifically for a different task (constructing dynamic checklists from bottom-up). We have updated Section 2 in the paper with a little more detailed discussion regarding the differences between the datasets and the tasks we are presenting in this paper (compared to [13, 14]). In the current revision of the paper, we also mention that we have translated the LICD data set to English.
>
> -Regarding the acronyms, We have also updated the paper to improve readability (especially Section 1 and the abstract). We also added more details to introductions to important concepts, such as sustainable development goals (SDG)

---

> ### Comment · Area_Chair_vNtm · 2022-08-28
> **Discussion period near closing**
>
> Dear Reviewer,
>
> Thanks for your review for the submission. The authors have provided a rebuttal. The discussion period is only one day from closing. Can you indicate how much the rebuttal has addressed your concerns, and provide your current evaluation (if changed)? Thank you in advance.
>
> Your AC

---

### Official Review · Reviewer_eGQ1 · 2022-07-27
**Labour Inspection dataset curated from inspections done for various industries in Norway during 2013 to 2019**

**Rating:** 5
**Confidence:** 4
**Clarity:** The paper is clear to read and unders…

**Strengths:**

If the ML research on this dataset can show practically useful results, then it can be very useful for agencies that perform an important job of making sure that the employees work in safe working conditions. ML based recommendations can help inspection agencies in shortlisting high risk companies to inspect on priority and in quickly creating the right checklist of questions to evaluate during inspections.

The authors promise to share dataset publicly after the paper review process. At this point, it is hard to comment on the accessibility of the dataset for wider research community. As per authors, the dataset is compliant to GDPR and Norwegian privacy protection laws. It is also made sure that the data does not contain privacy violating features by getting dataset reviewed by the management of inspection agency.

**Weaknesses:**

The paper as several weaknesses as listed below:

1. *Evaluation metrics:* The two category of experiments used to demonstrate the application of ML for predicting target variables are evaluated using metrics that dwell on hard labels, i.e. accuracy, precision and recall. The models used in experiments return soft scores and in order to convert soft scores to hard labels (0/1 or 0..n if multi class), a threshold is chosen as pivot point. The authors did not make it clear how threshold was chosen and in fact the pivot point is unknown. Seems like default threshold of 0.5 was chosen but 0.5 does not make sense for non calibrated models like GradientBoost. It is unknown, what is the desirable precision or desirable recall that can be used for tuning the threshold. Instead, for comparisons of different models, it would be better to first evaluate the models in terms of metrics that can use soft scores, like AUC (micro/macro) and accordingly show better performing models.
2. *Missing feature engineering details:* Missing values getting imputed by 0 limit ability for researcher to use their own imputation method. A researcher might want to impute with mean or median, or a -1 (depending on the range of values in the data). Was there any data transformation done on any features before passing to the models? How many features were categorical and how were they treated? Such details on feature engineering are missing.
3. *Feature Selection:* Much discussion is provided on feature selection but there is no result based on full set of features. In most of the chosen models, the use of regularization can inherently handle 575 features and remove ones that are less useful. As a result, feature selection step in many cases can turn out to be of minimal benefit. Also, if author believe feature selection to be an important step, then a discussion of top features impacting the target outcome is missing.
4. *Lack of error bars:* The authors mention use of cross validation but error bars are not provided in results. If cross validation was not used, then details about train, validation and test splits is missing. It would have been better to add details on how the sizes of train, validation and test sets if used.
5. *Lack of discussion on application of results:* The performance for both tasks using different models is low. Authors have not mentioned what is the target performance that could be of a practical help for agency. Is the current performance for both tasks practically useful? If yes, how does the agency plan to use the predictions? Moreover, CLSP is a multi class classification task and it is unclear which class is more predictable in the 10.6% accuracy achieved by DT model.
6. *Dataset not scalable to other countries limiting its impact:* The dataset is curated by Norwegian Inspection agency and comprises of inspections performed in Norway. This limits the impact and usefulness of dataset to broader research community from other geographies.
7. *Imbalanced vs. balanced*: 26%-74% class ratio is generally not required to be under/over sampled. Under/over sampling is generally done for highly imbalanced datasets (e.g. 1 in 1000 instances). Moreover, test sets are kept in original class ratio when under/over sampling is performed to provide real evaluation metrics that will mimic production scenario. Instead, if the class ratio is not 50-50, use of right evaluation metrics (like AUC instead of Accuracy) comes into play.
8. *Data distribution analysis:* It would be useful to provide distribution analysis of different attributes present in the data. How many attributes were categorical, how many numeric, what is the relationship of different variables with target variable etc.



**Additional Feedback:**

Please check following lines for corrections:

* Line 52: Mention January 2013 but in attached data set documentation, 1/1/2012
* What (other) tasks could the dataset be used for?:  “The data set could be relevant for benchmarking ML-methods dealing with imbalanced” → Disagree that 74/26 is the level of imbalance that can be used in research of imbalanced datasets. In real world scenarios, we deal with imbalanced datasets with 1/1000 or even 1/10000 class ratios.



**Correctness:**

The dataset is pulled from agency’s database that performs HSE inspections in Norway. The primary author work at the same agency and has done the management review to make sure dataset can be released publicly. The organizational information is taken from the time of inspection rather than most recent information, which makes sense for the task at hand.

However, the dataset creation does not seem 100% correct. For example, the authors themselves imputed missing values with 0 which should be left to the user of the dataset to decide how to impute missing values. Similarly, the choice of metrics and evaluation methodology of baseline ML approaches does not seem to be correct, as also mentioned in the weakness section.

**Documentation:**

Data is currently not hosted publicly, but the authors have mentioned that the datasets will be hosted after paper acceptance. Meta data includes category level information but lacks feature level information, or about what percentage of features are categorical vs. numeric vs. text.

**Ethics:**

The author has gained approval from agency management who owns the datasets. It is although not mentioned whether organizations under review are aware about this release or not, or whether they signed agreement to disclosure. Saying that, the authors have put efforts to not let the data reveal identity of any organization including removing small organizations, removing features that can violate privacy, and making sure that the dataset is GDPR and Norwegian privacy protection laws complaint.

**Relation To Prior Work:**

The related work in similar as well as different domains are clearly discussed.

**Summary And Contributions:**

The authors release a Labour Inspection Checklists dataset that contains ~63k instances of inspections conduced by Norwegian Labour Inspection Authority (NLIA) during the period of January 2013 to June 2019. The dataset contains 575 features and 2 target variables. The features contain meta data on organizational or financial information about the company under inspection. The target variables are 1) Checklist inspection topic, 2) Is company compliant for all questions or non compliant for at-least one questions in the checklist. The dataset is based on actual daily operations of NLIA without any transformations.  To show how the authors envision this dataset as a useful dataset for ML community, the results from baseline experiments for prediction of target variables are discussed.

The contributions from authors include curation of a dataset that can be used to make the inspections of HSE violations easier for governmental agencies that conduct labour inspections. ML can help such agencies with limited resources to better plan the inspections.

---

> ### Author Response · Authors · 2022-08-09
> **Many of the weaknesses can be addressed by updating the data set or the paper with more details.**
>
> Hi,
>
> Thank you for taking your time to review the paper. Regarding the weaknesses.
>
> 1.	Evaluation metrics: We used the default settings in the Sklearn framework for the experiments, which uses a decision function by default sets prediction thresholds to 0.5. While it is possible to use a correlation coefficient or ROC curve to set the threshold, we avoided to do so to not increase the complexity of the experiment. We agree that it should be mentioned more explicitly in the paper.
>
> 2.	Missing feature engineering details: We will update the data set so that it contains missing values. We will also update the paper on how features are handled.
>
> 3.	Feature selection: we will add a discussion regarding the top features. I think we somehow forgot to include this in the first place.
>
> 4.	Lack of error bars: We will add standard deviation from the cross validation to the paper
>
> 5.	Lack of discussion on application of results: Regarding the performance we would argue that the models would be useful, as they work better than making qualified guesses. However, we do not think the models are strong enough to be worth getting into a production state. However, the data set and the paper could promote more research into the area and lead to stronger machine learning models being developed.  We also plan to update the paper with a discussion regarding this.
>
> 6.	Dataset not scalable to other countries: We wish it were possible to have data from multiple authorities, however many authorities in other countries today don’t collect data that is useable for machine learning (cf. Related Work section in our paper). We think and hope our data set could lead to more published research into using machine learning for labour inspection tasks. Ultimately, more interest and research into the topic may also lead to more data being collected as a result.
>
> 7.	Imbalanced vs. balanced: When we conducted the experiments, we tried to balance the data set after making observations in the experimental results that suggested class imbalance. One of the observations was that some of the machine learning methods produced consistently high prediction scores for all test cases (leading to 100% predicted positives). In our experiment we found that the test scores in overall improved after balancing the data set*.  Thus, we believe that this supports the decision to balance the data set. However, we can “tone down” the focus on data set imbalance in the paper a bit (as suggested under additional feedback).
>
> 8.	Data distribution analysis: We will update the paper with a discussion of this.
>
> \*We used balanced accuracy to compare the results between the imbalanced and balanced version of the data set, which uses sampling weights to adjust for class imbalance.  See https://scikit-learn.org/stable/modules/model_evaluation.html#balanced-accuracy-score

---

### Official Review · Reviewer_vyep · 2022-07-28
**Danish-language dataset on labor law compliance**

**Rating:** 8
**Confidence:** 3

**Strengths:**

The manuscript illustrates that one contribution of this dataset it “expands on previous SDG related NeurIPS datasets and benchmarks (such as SustainBench),” since none of those datasets cover SDG 8. This is a notoriously difficult arena to get good data on (especially financial information of regulated entities), and I laud their efforts to make this information public.


**Weaknesses:**

This dataset is focused in on one country only. This is still an advance, as I appreciate this data is difficult to obtain, and regulations vary from place to place that will characterize what is in “violation” or not differently. However, one major difference between this, and, for example SustainBench that they mention in related work, is that the latter seeks to at least keep track of more than one country.

While they contextualize this dataset within other related work, it’s not clear to me that all of the citations refer to datasets (some appear to refer to manuscripts that illustrate analysis, others to datasets that may be more or less challenging to parse).

In an effort for harm prevention, the dataset that is publicly available strips out information such as geography, violation type, or facility name that may actually prove useful for predictions (e.g., in the case of spatial spillovers). Perhaps there’s a way the authors can indicate a ‘trusted analyst’ position for others to be vetted to use the full dataset so as to incorporate such data a la Census Sworn Status? It’s also surprising to see that 74% of inspections yield violations, and that makes me wonder if there’s degrees of variation in compliance with inspections that one may want to take into account in subsequent analyses, if sufficient information is available, and/or if there are ways to subdivide/focus on a subset of the population.

The linked dataset is also in Danish, which limits accessibility (there is an option in the upper right to convert to English but then it no longer remains on the same page. An English language link, if available, would enhance accessibility.  Else a note indicating the webpage is not in English would also be helpful.


**Additional Feedback:**

I'd invest in this paper by citing it and would be curious to see what further analysis results. Other countries have similar types of questions and problems, so having some model to draw upon and an understanding of which factors may be most strongly associated with violations could help guide their inquiries even when regulations vary.

**Clarity:**

The paper is well written and easy to read. Figure 1 is also very useful for guiding the reader, as are the two possible use cases.

**Correctness:**

The largest challenges that seems to be plaguing their analysis is lack of variation in the dependent variable (e.g. 75% of the observations have some form of noncompliance). I do wonder if considering the distribution of compliance problems would yield more interesting and useful insights.

**Documentation:**

As mentioned above, the linked dataset (in a footnote) is in Danish, which limits accessibility (there is an option in the upper right to convert to English but then it no longer remains on the same page. An English language link, if available, would enhance accessibility.  Else a note indicating the webpage is not in English would also be helpful.  Unclear what the hosting, licensing, and maintenance plan is.

**Ethics:**

Whereas the authors say they make identification of company names difficult, one might actually believe we have an imperative *to* reveal the names of companies with labor violations rather than aggregate them/hide them. The harm prevention seems to be interpreted as harm to the companies whereas the harms to people, the public, and laborers for bad health and safety practices also seems relevant to account for.

**Relation To Prior Work:**


This work is contextualized and noted as different form other works in the content application (health and safety instead of food inspections, crime, or environmental enforcement).


**Summary And Contributions:**

This work seeks to offer a new dataset on over 60,000 workplace inspections as well as completed checklists related to labor law compliance in Norway between 2012 and the beginning over 2019. This dataset is intended to help identify patterns of health, safety, and environmental violations using ML techniques as well as help determine which are the appropriate checklists to use in a given context. The authors note that the sample experiments that they show using this dataset unfortunately do not exhibit a high performance with respect to predicting violations but are optimistic that novel methods using this data may be able to outperform their benchmark experiments.

---

> ### Author Response · Authors · 2022-08-09
> **A little misunderstanding regarding the location and origin of the data set**
>
> Hi, thank you for your time and insightful review. We think that there is a little misunderstanding regarding the data set. We have included both the balanced and unbalanced version of our proposed Norwegian LICD data set in the Supplementary materials (the link is located on this page). We have also translated our LICD data set to English to make it readable and understandable for a broad audience.
>
> The link in the paper is to the Danish Labour Inspection Authority’s Smiley Data set, which we mention as related work. Compared to our proposed data set, the Danish Smiley data set lacks complete records (with checklists). There are also no complete records with negative labels. The Danish Smiley data set is also only available in Danish.
>
> We apologize for the misunderstanding and we have edited the paper (Section 2) in an attempt to distinguish our Norwegian LICD data set from the Danish Smiley Data set more clearly.

---

> ### Comment · Area_Chair_vNtm · 2022-08-28
> **Discussion period near end**
>
> Dear Reviewer,
>
> Thanks for your review for the submission. The authors have provided a rebuttal. The discussion period is only one day from closing. Can you indicate how much the rebuttal has addressed your concerns, and provide your current evaluation (if changed)? Thank you in advance.
>
> Your AC

---

### Official Review · Reviewer_irin · 2022-07-28
**A very important topic and dataset for protecting labor rights and safe working environments. Additional information is needed.**

**Rating:** 7
**Confidence:** 3
**Clarity:** The paper is well written, but some d…

**Strengths:**

This new and unique dataset is addressing a very important labor safety issue, promoting a safe working environment for all workers and motivating further research on this subject.
The authors conducted initial features evaluation and basic data visualization to highlight how some of the key features of the dataset are distributed.
The high number of relevant checklists used within a single industry code points to the fact that there is significant diversity in the health, environmental, and safety risks for organizations, even within the same industry.

Good job on being specific about the equipment used: “A Dell Precision 5560 laptop with Intel i9 11950h at 5Ghz, 64GB RAM at 3200Mhz, Nvidia Quadro RTX A2000, and Windows 10 are used for the experiments.”

The full dataset was imbalanced in terms of non-compliance. To address this issue, the authors created a balanced set for ML models.

**Weaknesses:**


Several questions were not fully addressed in the manuscript:

*How was the data balancing created? Which samples were selected?
*Were there any work done for the analysis of the similarities between the users and industries?
*Do the industry code numbers also maintain the similarity information between the industries? Are industries that have numbers that are closer also perform more similar business rather than distant codes? What could be the reason for the spike gaps between codes in Figure 2a for example, between 50 and 60, which correspond to most of the building and road construction industries? Could it be better to treat them as categorical features rather than numeric?

*Sequential Selection was unable to complete within two hours for any of the larger feature sets. Would running the selection longer help?

The results for the trained models on the unbalanced LICD have room for better performance and highlight the necessity for running the experiment on a balanced dataset or more complex models. However, the results on the balanced data improved also leave room for improvement. Details about the GridSearchCV hyperparameter tuning were not specified in the text. Thus, better results could potentially be achieved using a broader or better hyperparameters tuning mechanism in future work.

The variance for the cross-validation runs is not shown in the manuscript. Could you please add it?


**Additional Feedback:**


Thank you very much for working on such an important topic and creating a new dataset and bringing attention to the safety of work environments.



**Correctness:**


The main claim about the importance of the dataset seems to be accurate. The authors provided a broad set of models that could be used as well as feature selection mechanisms.

It was mentioned that each configuration is evaluated using 5-fold cross-validation with randomly selected training evaluation sets from LICD. The performance is measured in terms of balanced accuracy, accuracy, precision, and recall scores using the available methods in the Scikit-learn library. The results for each method are recorded by calculating the average of the scores reported from the 8 feature selection configurations. However, the variance was not found in the main text.

The authors were upfront and clarified: “All in all, none of the methods above perform well in terms of classification performance on either LICD or LICD/B. For this domain, a high precision score is important because it means that a higher percentage of non-compliance is found on average per inspection. A high recall is also desirable, but less important as long as there are a sufficient number of predicted positives. A decent accuracy is also important, in order to ensure that each method has decent classification performance.”


**Documentation:**

The dataset was well described in the manuscript at each step starting from the inspection, extraction, and the outcome structure as well as supplementary materials. The link to the data was provided, however, I was not able to get access to the data using the links. Could a direct link to the data and documentation be provided?


**Ethics:**

The authors write that the “LICD contains information from sources that are publicly available, to some extent. Some of the organisational and financial features are available through Norway’s official Central Coordinating Register, but the register doesn’t contain historical records.The checklists and inspection outcomes are only available from inspection reports. Some of these may be available to the public, but access is granted only on a case-by-case basis via requests.” This means that the created and published dataset would allow access to this information to the whole community.

Preserving the anonymity of the assessed organizations is a very challenging issue that has to be a priority when developing the dataset. The authors declare that their strategy for harm prevention was to ensure that the dataset only contains information that is safe to make publicly accessible. “For this reason, we have not included details such as which regulations were found to be non-compliant. As an extra precaution we also make it difficult to identify organizations in the dataset. To do so we have excluded the organisation names and identifiers from the dataset. We have also excluded the precise location of the organisations for the same reason.”
I believe that it is important to conduct additional convincing work on making sure that the privacy of the organizations is insured. For example, trying to deanonymize the dataset.

Additionally, the data might be potentially biased towards the inspections available, however, it seems to be a reasonable start. It would be great to have a discussion about it as well.

We have to be careful about the anonymity, consent, and biases in the existing samples.


**Relation To Prior Work:**

Prior work was described in relation to the inspection domains, existing data, and ML models.


**Summary And Contributions:**

The paper introduces a new labor inspection checklists dataset (LICD) for selecting relevant checklists to address working environment violations in organizations, which is very important for protecting labor rights. LICD consists of 63634 instances with past inspections and contains 575 features and 2 possible target variables: Non-compliance and checklists.
Based on the target variables, the dataset could potentially be used to select a relevant checklist to (1) survey a given target organization as well as to (2) classify whether non-compliance is found at an inspection, for a given organization and checklist. It could be used by inspectors to select high-risk organizations for inspections.
The authors used feature selection in the experiments with Anova F, χ 2, coefficient importance from model training, Mutual information, sequential selection (forward search), and recursive elimination. The ML methods used for the experiments were Decision tree (DT), Logistic regression (LR), Naive Bayes’ Classifier (NBC), K-Nearest-Neighbor (k-NN), AdaBoost, Gradient Boost, and Multi-layered Perceptron (MLP). GridSearchCV for hyperparameter tuning for k-NN, AdaBoost, GradientBoost, and MLP. Each ML method was evaluated on 8 different feature set sizes that are selected via feature selection. There are only minor differences between most of the methods in the test. Sequential Selection, Anova F and Mutual Information have the best-recorded accuracy scores.

Future work using this dataset could, for example, involve a multi-objective optimization problem to select the most relevant inspection checklist that maximizes the number of violations found in a given organization.

---

> ### Author Response · Authors · 2022-08-09
> **We will update the paper with additional information address the questions.**
>
> Hi, thank you very much for taking your time to review the paper. We will update the paper to address your remarks/questions and provide answers to some of the questions here:
>
> In short, data balancing was done via random undersampling before cross-validations (split). We kept all the observations with negative labels and randomly selected a number of records with positive labels, so that there is an equal number of observations with positive and negative labels in the final balanced data set.
>
> Industry code numbers can maintain some degree of similarity between different industries. However, we did not address this in the paper due to space/scope restrictions. Formally we treat industry codes as a categorical feature (one-hot vector) in the paper. However, we think that exploring other ways process features could be a good direction for future work.  Looking at the paper, we realized that we forgot to mention how we prepared/encoded the features for the experiments. We have added this information as a new subsection in Section 6 of the paper in the current revised version (see Section 6.1).
>
>  We used a simple hyper parameter tuning method for the paper in order to keep the details and setup of the experiment simple. We plan to add more details about the hyper parameter tuning to the paper (future revision).
>
> We have also updated the paper with more discussions regarding performance. This also includes a discussion of how balancing the data set affects performance both from a machine learning perspective and from a real-world usability perspective. We also agree with the reviewer that there is likely possible to improve the results by introducing more complex models or by other measures such as transformations (feature extraction) or more complex feature selection methods. We have updated the paper with some more discussions of performance (Section 6.5). We have added a discussion in Section 6 (6.5) regarding the computational time/suggestions for solution to Sequential selection to also address your question about running time for this method.
>
> We have also included standard deviation for the experiments in Section 6 in the current revised version of the paper.

---

> ### Comment · Area_Chair_vNtm · 2022-08-28
> **Discussion period near closing**
>
> Dear Reviewer,
>
> Thanks for your review for the submission. The authors have provided a rebuttal. The discussion period is only one day from closing. Can you indicate how much the rebuttal has addressed your concerns, and provide your current evaluation (if changed)? Thank you in advance.
>
> Your AC

---

### Official Review · Reviewer_jevu · 2022-07-30
**The datasets is not informative enough**

**Rating:** 5
**Confidence:** 4
**Correctness:** The claims seem correct to me.

**Strengths:**

1. The authors are releasing a dataset on a relatively unexplored task; however, it could be helpful for better inspection by agencies.
2. The dataset consists of 63634 inspections which is to some extent large.
3. Several experiments are conducted among two tasks, predicting checklists and non-compliance with different machine learning models.

**Weaknesses:**

1. The whole data is only coming from a single authority (NLIA). Therefore it might not be directly useful for other parts of the world.
2. The performance of the trained machine learning models is extremely poor. That probably could be because the data is not informative enough for the defined task. Poor performance sometimes means that the data is not informative enough, which I think is the case here.
3. The authors do not elaborate on features that are provided for organizations and why they believe they are informative enough. Also, for the checklist, I think there is no single ground truth for it. The fact that a set of checklists are selected during the inspection by the inspector does not mean that the chosen set is optimal, making predicting the list even harder.


**Additional Feedback:**

I think in the current format the data is not informative enough for the task that the authors have in mind. Changing the input features or the desired output can help solve this issue.

**Clarity:**

The paper is not well written in my opinion. I think the authors can do a better job presenting sample input (features) and outputs (checklist) in the paper for their defined task.

**Documentation:**

Additional details about the feature selection and potential checklists are missing.

**Ethics:**

Some organizations might be recognizable based on the features that are provided., especially since all the organizations are from a small region. The authors excluded organizations with less than six employees, but it does to completely solve the issue.


**Relation To Prior Work:**

The authors mention previous datasets in this domain such as "Machine learning for environmental monitoring" by Hino et. al.


**Summary And Contributions:**

This paper introduces The Labour Inspection Checklists Dataset consists of 63634 instances where each instance is an inspection conducted by the Norwegian Labour Inspection Authority. 575 features are provided for each organization that is inspected and potential target variables are checklists (list of items that should be inspected for the organization) and non-compliance. The authors experiment with several machine learning methods on their task, but the performance for all of them is very poor.

I appreciate the authors' response, and I have updated my score. However, I still believe that the extremely poor performance of ML models on this dataset is a red flag. The dataset in the current format is probably not informative enough, which can even discourage researchers from working on this area as the trained models don't generalize. I would encourage the authors to include other sources of data if possible to mitigate this issue.

---

> ### Author Response · Authors · 2022-08-09
> **Need more research to understand what information (features) should be collected for machine learning in labour inspection tasks**
>
> Hi, thank you for taking your time to review the paper. We think that you had some good points regarding the weaknesses of our paper that we would like to discuss:
>
> Weakness 1: We wish it were possible to have data from multiple authorities, however many authorities in other countries today don’t collect data that is useable for machine learning (cf. Related Work section in our paper). As mentioned in the paper, there have also not been much published research into how machine learning could be used for labour inspection tasks. We think and hope our data set could lead to more research into this topic. Ultimately, more interest and research into the topic may also lead to more data being collected as a result.
>
> Weakness 2:   We think you have a point regarding the informativeness in the data and that higher performance potentially could be achieved by collecting more data. However, our current data set consist of all (Non-harmful, GDPR and ethically compliant) information the Norwegian Labour Inspection Authority has available. To get a more informative dataset and boost Machine learning performance, we first need more research to understand what kind of data that should be collected. We believe that our paper and data set could be a good starting point for this.
> We updated Section 6 in the paper with a new subsection that discusses the performance of the machine learning models. In this discussion we also mention the need for more knowledge regarding feature/data collection.
>
> Weakness 3:  An inspector selects only a single checklist for each inspection so there is only one ground truth per observation (inspection). If we have formulated any definitions in the paper wrong, please let us know. We have done some minor changes to the problem definitions in Section 1, in an attempt to improve the language and clarity. Regarding the features for the organizations, we have updated the Section 6 with a discussion regarding the highest ranked features that were selected in the experiments. This should partly address the concerns about the informativeness of the features in the dataset and has also been requested by other reviewers. We have also updated the paper with a discussion of whether multiple optimal checklists could exist and the possibility for adapting the machine learning problems accordingly. However, we think that empirical trails and/or analysis are needed in order to find out whether or not multiple optimal checklists are common for labour inspections. We also plan to add a table with examples to highlight what information some of the more important features contain.
>
> Regarding ethics: We are aware of the possibility of recognizing an organization based on sets of features.  The information organisations are taken from the inspection date (historical record) and is therefore difficult to match with any current public register. This is done as an extra precaution as we don’t consider the information in the data set to be harmful. Inspection results are generally accessible on individual case-by-case-basis via freedom of information act requests. We have also excluded any inspections in organisations with less than 6 employees from the data set, to ensure that there are no privacy/GDPR issues.

---

> ### Comment · Area_Chair_vNtm · 2022-08-28
> **Discussion period near closing**
>
> Dear Reviewer,
>
> Thanks for your review for the submission. The authors have provided a rebuttal. The discussion period is only one day from closing. Can you indicate how much the rebuttal has addressed your concerns, and provide your current evaluation (if changed)? Thank you in advance.
>
> Your AC

---

### Author Response · Authors · 2022-08-26
**Revision of the paper**

We have revised the paper to address the reviewers' concerns and questions.  Some of the most improvements includes:

-We ran the experiments again and update the paper with standard deviations from cross-validations. We also included AUC for the NCP experiment. The experiment on the balanced data set is done by balancing only the training folds while keeping the validation folds unbalanced (as requested by reviewers). We have also updated the results and discussions in Section 6 to reflect the changes to the experimental setup.

-Discussions of the most important features in the data set selected by the feature selection algorithms that we used. This also implicitly include a discussion about informativeness of the selected features. Rationale for using feature selection is described in the beginning of Section 6.

-Discussion about the performance of the tested ML methods from both a ML performance and a practical (real-world usability) perspective.

-Improved abbreviations and discussions about previous work.

-Rationale for balancing the data set. We plan to conduct a new experiment for NCP where we adjust prediction thresholds rather than balancing the data set.

-Added information about data preprocessing. We also listed types (categorical, binary etc) for the features that are discussed in Section 3.

-Updated LICD and LICD/B datasets so that they no longer have their missing values replaced with 0.

-We have also made many other minor changes to improve the paper, datasets and the experiments according to the feedbacks. As a result, the paper now consists of 10 full pages excluding references.

We will continue to improve the paper as much as possible after the discussion deadline.

---

### Comment · Reviewer_eQM2 · 2022-08-29
**brief, 11th-hour ethics review**

This kind of dataset is a valuable contribution, and authors' thoughtful approach to assembling and constructing it is admirable. There are no perfect choices when constructing or employing a training data set, and the sampling methods and de-identification choices (particularly the exclusion of small organizations) used here are thoughtful and well-designed to minimize harm.

---

### Meta-Review · Area_Chair_vNtm · 2022-09-10

**Recommendation:** Accept
**Confidence:** 4

**Metareview:**

The paper proposes a dataset on labor inspections, where a set of curated features are used to predict inspection checklists and violations. The perspective is unique and fresh to the machine learning community, and the underlying goal of safe working environment is relevant to everyone. The reviewers mostly agree with this viewpoint, despite some pointed out the current results are not competent or some details/analysis are missing. This leaves room for future research, and the main contribution is the hard-to-obtain dataset and the new problem. The author also provided detailed rebuttals with a revised draft, addressing many review comments. In my opinion the dataset/problem contribution outweighs potential flaws in experiments, and I recommend the paper to be accepted.

---

### Decision · Program_Chairs · 2022-09-16

Accept